# Advances in the Oral Administration of Somatostatin Receptor Ligands in Acromegaly: A Systematic Review Focusing on Biochemical Response

**DOI:** 10.3390/pharmaceutics16111357

**Published:** 2024-10-24

**Authors:** Clémence Reverdiau, Damien Denimal

**Affiliations:** 1Department of Pharmacy, Faculté des Sciences de Santé, Université de Bourgogne, F-21000 Dijon, France; 2Department of Clinical Biochemistry, Centre Hospitalier Universitaire Dijon-Bourgogne, F-21000 Dijon, France; 3Center for Translational and Molecular Medicine, Université de Bourgogne, F-21000 Dijon, France

**Keywords:** octreotide, paltusotine, acromegaly, oral route

## Abstract

Recent advances in pharmaceutical technology, aimed at overcoming poor drug permeation across the intestinal–epithelial membrane and the challenges posed by the acidic gastrointestinal environment, have led to the development of orally administered somatostatin receptor ligands (SRLs). This development represents a promising step forward in the management of acromegaly, offering an alternative to the limitations associated with injectable SRLs. Several key clinical findings have emerged in the past two years, notably including the results from the extension phase of the MPOWERED trial, which evaluated oral octreotide capsules (OOCs), and the placebo-controlled PATHFNDR-1 trial using paltusotine. This prompted us to conduct a systematic review of the literature focusing on the efficacy of oral SRLs in controlling acromegaly, based on biochemical response. Of the 136 reports identified through our search on Medline and ClinicalTrials.gov, twelve were included, encompassing data from five interventional trials. Both OOCs and paltusotine demonstrated the ability to maintain biochemical control in patients previously controlled with injectable SRLs. While long-term maintenance was confirmed for OOCs, no data are yet available for paltusotine. Several gaps remain, such as the need for head-to-head comparisons between OOCs and paltusotine, and clinical trials in patients who have not received prior injectable SRL treatment.

## 1. Introduction

Acromegaly is characterized by an excessive production of growth hormone (GH) by the anterior pituitary gland. The prevalence of acromegaly is estimated at 5.9 cases per 100,000 individuals, with an incidence rate of approximately 0.38 cases per 100,000 person-years [1]. Medical therapy is recommended for patients with persistent disease despite surgical resection of the adenoma as well as for patients in whom surgery is not appropriate [2]. The long-acting formulations of the somatostatin receptor ligands (SRLs) octreotide and lanreotide are currently recommended as first-line medical therapy, while long-acting pasireotide could be considered as a second-line treatment [2]. SRLs bind to somatostatin receptors and mimic their physiological effects, leading to a reduction in GH secretion and subsequently insulin growth factor (IGF)-1 secretion by the liver.

Octreotide, lanreotide, and pasireotide are SRLs of a peptide nature, initially developed in injectable formulations due to their susceptibility to enzymatic degradation and low epithelial permeability in the intestine. They are administered through monthly intramuscular or subcutaneous injections of long-acting formulations. Injectable SRLs (iSRLs) exhibit numerous disadvantages such as injection site pain, nodules, bruising, inflammation, and scarring [3,4,5]. A real-world survey showed that 70% of patients reported experiencing pain at the injection site lasting up to a week, with other common adverse effects including nodules (38%), swelling (28%), bruising (16%), scar tissue formation (8%), and inflammation (7%) [3]. In addition, nurses have only a small amount of confidence that the octreotide syringe will not be clogged and will be somewhat easy to use during injection [6].

The development of oral formulations of SRLs has raised hope in mitigating these adverse effects and increasing compliance and quality of life [7]. Significant advances in oral formulations of SRLs have been made in the last decade. In particular, oral octreotide capsules (OOCs) have been developed through the enteric coating of capsules with a pH-dependent methacrylic acid copolymer preventing chemical and enzymatic degradation, but above all through the development of absorption enhancers to overcome low epithelial permeability [8]. In particular, the Transient Permeation Enhancer^®^ technology (TPE^®^, Chiasma™, Jerusalem, Israel) has been used for the development of OOCs [9]. TPE^®^ consists of an oily suspension of the peptide drug, sodium caprylate (C8), glyceryl monocaprylate and tricaprylate, and polyvinylpyrrolidone. C8 plays the role of a permeation enhancer by causing a transient opening of the intestinal epithelial tight junctions, thus creating a paracellular pathway for peptides with a molecular weight lower than 10 kDa (octreotide: 1.14 kDa) [10].

The pharmacokinetic parameters, such as peak plasma concentration and area under the curve, are similar after oral administration of 20 mg octreotide or subcutaneous administration of 100 µg octreotide in human subjects [11]. The first results of phase III trials for OOCs were published in 2015 through the open-label CH-ACM-01 trial [12], followed by the results of the randomized controlled treatment OPTIMAL trial in 2020 [13]. Subsequently, Mycapssa^®^ (OOC) received regulatory approval from the US Food and Drug Administration (FDA) in June 2020 for the long-term maintenance and treatment of patients with acromegaly who have responded to and tolerated treatment with injectable octreotide or lanreotide. The European Medicines Agency delivered a marketing authorization for Mycapssa^®^ in December 2022.

In addition to OOC, paltusotine, a novel oral SRL, has been recently developed by Crinetics Pharmaceuticals™ [14]. Paltusotine is original among oral SRLs due to its nonpeptide nature (Figure 1) and its high specificity for the somatostatin receptor subtype 2, while octreotide has an affinity for both somatostatin receptor subtype 2 and subtype 5. The placebo-controlled phase II ACROBAT Evolve trial (NCT03792555) was designed to assess the safety and efficacy of paltusotine in patients that are responders to long-acting iSRLs, but the results were not published due to the positive results of the ACROBAT Edge trial (NCT03789656). This open-label, phase II, single-arm study demonstrated satisfactory maintenance of biochemical response, leading the FDA to grant orphan drug designation to paltusotine for the treatment of acromegaly. Very interestingly, the first results of the placebo-controlled phase III PATHFNDR-1 trial (NCT04837040) evaluating oral paltusotine in patients with acromegaly previously controlled with iSRLs were recently published in June 2024 [15].

This prompted us to conduct a systematic review of clinical trials of SRLs administrated via the oral route in patients with acromegaly by focusing on their efficacy based on the biochemical response, which is the strongest predictor of patient outcomes.

## 2. Materials and Methods

Our systematic review was presented according to the PRISMA 2020 statement [17].

### 2.1. Eligibility Criteria

Table 1 shows the inclusion and exclusion criteria used to select the clinical trials for the review, according to the Population, Intervention, Comparison, Outcome, Study (PICOS) format.

### 2.2. Information Sources and Search Strategy

We conducted a search procedure using the PubMed/MEDLINE and ClinicalTrials.gov databases. The search strategy was based on the following words: “acromegaly”, “octreotide”, “lanreotide”, “pasireotide”, “paltusotine”, and “oral”. In addition, the references cited in the included studies were examined to select additional reports for screening. All reports published up to 15 June 2024 were included for further screening.

### 2.3. Selection and Data Collection Process

The two authors independently screened the titles and abstracts, excluding reports and studies based on the predefined inclusion and exclusion criteria. The remaining studies were integrally read to definitively judge on their inclusion. Disagreements on the eligibility of studies were resolved through discussion and consensus between the two authors. Data from each report were independently collected in Excel sheets by the two authors. No automation tool was used in the selection process or in data collection.

### 2.4. Data Items (Outcomes)

The primary outcome was the proportion of responders at the end of the treatment period, as determined by serum IGF-1 and GH levels.

### 2.5. Study Risk of Bias Assessment

The risk of bias in the included studies was independently evaluated by the two authors using the revised Cochrane risk-of-bias tool for randomized trials (RoB 2) and the ROBINS-I tool for uncontrolled trials [18]. Any discrepancies in judgment between the two authors were resolved through discussion.

### 2.6. Effect Measures

The effect measure of the primary outcome was the percentage of responders among enrolled participants.

### 2.7. Synthesis Methods

When the 95% confidence intervals of the proportion of responders were not available in the reports, they were determined using the modified Wald method (GraphPad Prism, version 9.5.0).

## 3. Results

### 3.1. Study Selection

The flowchart in Figure 2 illustrates the selection process for the studies included in our systematic review. Initially, 136 reports were identified through our search strategy. The majority (*n* = 87, 64.0%) were excluded based on the initial screening of their titles and abstracts due to non-adherence to our inclusion criteria. An additional fourteen records (10.3%) were excluded either due to non-retrieval or because the biochemical results were not yet available. The unpublished biochemical results were related to the following trials: ACROBAT Evolve (NCT03792555) [19,20], ACROBAT Advance (NCT04261712) [21], and PATHFNDR-2 (NCT05192382) [22].

Of the nineteen studies assessed for eligibility, five were excluded since they were literature reviews, one which was a preclinical study on animals [23], and another one involving a study population of healthy volunteers [24]. The interim results from the single-arm ACROBAT Advance study presented at the Endocrine Society’s annual meeting ENDO 2023 in June 2023 were not included in our review due to the lack of a peer-review process [25]. For the same reason, we did not include the results from the double-blind placebo-controlled PATHFNDR-2 study presented at ENDO 2024 in June 2024 [25].

Ultimately, we included twelve reports from the following five clinical trials: CH-ACM-01 (NCT01412424), OPTIMAL (NCT03252353), MPOWERED (NCT02685709), ACROBAT Edge (NCT03789656), and PATHFNDR-1 (NCT04837040). Table 2 and Table 3 summarize the key characteristics of these five interventional trials. Three of these trials assessed the efficacy of OOCs (CH-ACM-01, OPTIMAL, and MPOWERED), while the remaining two evaluated oral paltusotine (ACROBAT Edge and PATHFNDR-1).

All the selected trials were designed to include a core phase followed by a single-arm extension phase. The core phase consisted of open-label single arm trials (CH-ACM-01 and ACROBAT Edge), double-blind placebo-controlled trials (OPTIMAL and PATHFNDR-1), and a double-blind iSRL-controlled trial (MPOWERED). A total of 458 individuals with acromegaly, all of whom had achieved biochemical control on iSRLs, were enrolled in the core phases of these trials. Among them, 234 received OOCs, and 77 received paltusotine. The extension phases of these trials were structured as open-label single-arm studies, with biochemical results available for CH-ACM-01, OPTIMAL, and MPOWERED. The extension phases evaluating paltusotine are currently ongoing.

Biochemical measurements of serum IGF-I and GH levels were conducted using the iSYS analyzer with dedicated reagents (ImmunoDiagnostic Systems™, Boldon, UK) across all five clinical trials.

### 3.2. Risk of Bias

Table 4 and Table 5 present the assessment of the risk of bias in the randomized and uncontrolled interventional trials, respectively. Several potential biases were identified. In the CH-ACM-01 trial, at baseline in both the core and extension phases, 11% and 17% of participants, respectively, had IGF-1 levels exceeding 1.3 × ULN, despite having IGF-1 levels below this threshold at the time of inclusion. In the open-label MPOWERED trial, six participants were randomly assigned to the OOC group with IGF-I levels greater than 1.3 × ULN, whereas none in the iSRL group had IGF-1 levels above this threshold. Additionally, patients receiving the OOC had higher pre-inclusion doses of iSRLs. Moreover, the proportion of participants in the MPOWERED trial achieving IGF-1 levels below 1.0 times ULN at the end of the OOC treatment was not reported. In the PATHFNDR-1 trial, there were baseline imbalances between the paltusotine and placebo groups in terms of sex ratio, disease duration, and previous injectable octreotide dosage.

### 3.3. Biochemical Response

Table 6 summarizes the rate of responders based on IGF-1 and GH levels in the five selected interventional trials. In the single-arm CH-ACM-01 trial, the rate of responders (i.e., IGF-1 < 1.3 × ULN) reduced from 91.4% at baseline to 64.2% after a 7-month treatment with OOC [12]. The reduction in the proportion of responders was only 6.3% during the extension phase.

In the placebo-controlled OPTIMAL trial, all participants had IGF-1 levels lower than 1.3 × ULN at inclusion but this reduced to 71.4% after 9 months with OOC treatment. The OOC had higher efficacy than placebo in maintaining IGF-1 levels below 1.0 × ULN (+62.7%). The responder rate was 92.6% (78.7–100%) during the 11-month extension phase with OOC.

In the iSRL-controlled MPOWERED trial, the rate of participants with IGF-1 levels inferior to 1.3 × ULN remained stable after 8 months of OOC treatment. The efficacy was not inferior to that of iSRLs. It should be noted that only responders to the OOC during the run-in phase were enrolled in the randomized trial.

In the ACROBAT Edge trial, IGF-1 levels were expressed in concentrations rather than in percentage of responders. Nevertheless, the comparison between the levels at baseline and after paltusotine showed that monotherapy with paltusotine for 13 weeks maintained IGF-1 levels in patients previously on iSRL monotherapy.

Lastly, the core phase of the PATHFNDR-1 trial revealed that oral paltusotine for 8 months maintained IGF-1 levels below 1 × ULN in 83.3% of patients in comparison to 3.6% in the placebo group.

## 4. Discussion

The oral administration of SRLs, such as OOCs and oral paltusotine, represents an active area of research aiming to provide an efficient alternative to iSRLs in the management of acromegaly. It represents a genuine hope to improve the quality of life of patients with acromegaly. This development represents a promising step forward in the management of acromegaly, offering an alternative to the limitations associated with iSRLs. In particular, iSRLs exhibit disadvantages such as injection-site pain, nodules, swelling, bruising, inflammation, and scarring. Additionally, some patients may experience worsening symptoms of acromegaly and higher IGF-1 levels close to the date of their next injection [5,35]. However, a recent survey revealed that 65% of patients preferred subcutaneous injections administered once every fourth weeks using a pen at home, compared to oral capsules taken twice daily, suggesting that the required three-hour fasting period remains a significant burden [36]. The transition from injectable to oral SRLs should not compromise disease control. Our review indicates that the current data are reassuring in this regard.

The most studied formulation is OOCs, administered twice daily in a fasting state. The pharmacokinetic parameters in human subjects are similar after oral administration of 20 mg octreotide or subcutaneous administration of 100 µg octreotide, particularly regarding the peak plasma concentration and the area under the curve [11]. The mean of the apparent steady state elimination half-life ranged from 3.19 ± 1.07 h on 40 mg of oral octreotide, to 4.47 ± 2.02 h on 80 mg [12]. The first phase III trial, CH-ACM-01, demonstrated a biochemical response after seven months of treatment in approximately two-thirds of patients previously controlled by iSRLs. A similar magnitude of biochemical response was observed after nine months in the OPTIMAL trial, which also showed a clear superiority of the OOC compared to the placebo. Notably, the degree of baseline control on iSRLs was predictive of subsequent biochemical response to the OOC. In CH-ACM-01, the proportion of responders to the OOC reached 84.5% in patients with baseline IGF-1 levels below 1.0 × ULN, compared to 61.6% in the overall study population. The MPOWERED study was the first head-to-head comparison of an OOC with iSRLs in patients previously controlled with iSRLs. It demonstrated the non-inferiority of OOCs to iSRLs after an 8-month treatment period. In addition, it showed that the biochemical response remained satisfactory over three years of maintenance therapy.

In contrast to OOCs, paltusotine offers the advantage of once-daily administration. Paltusotine is associated with increased plasma concentrations to doses up to 40 mg, and is eliminated with a half-life of approximately 30 h [24]. Preclinical studies have demonstrated that paltusotine, an agonist of somatostatin receptor subtype 2, dose-dependently suppresses recombinant GH-releasing hormone-stimulated GH secretion in rats, and dose-dependently reduces IGF-1 levels [37]. In clinical trials conducted in humans, the ACROBAT Edge trial demonstrated that switching from iSRLs to oral paltusotine for thirteen weeks did not compromise biochemical control in patients previously well-managed on iSRL monotherapy. Recently published data from the PATHFNDR-1 trial further confirmed that paltusotine can maintain the biochemical response in over 80% of patients previously controlled with iSRLs, even after eight months of treatment.

Several gaps remain in the clinical validation of oral SRLs, including the need for direct head-to-head comparisons between OOCs and paltusotine. Furthermore, no peer-reviewed data are currently available on the efficacy of oral SRLs in patients who have not received prior treatment with injectable SRLs. However, the ongoing double-blind, placebo-controlled PATHFNDR-2 trial (NCT05192382) is assessing paltusotine in patients who were either treatment-naïve or previously treated but had discontinued medications for at least four months. Preliminary findings from PATHFNDR-2, presented at the ENDO 2024 meeting, reported that, after 24 weeks of treatment, 42.5% of patients in the paltusotine group achieved serum IGF-1 levels below 1.0 × ULN, compared to only 2.4% in the placebo group [22]. Finally, data on the long-term efficacy of paltusotine are still pending. The open-label phase II ACROBAT Advance trial (NCT04261712) is currently ongoing and is designed to evaluate the safety and efficacy of paltusotine over one year of treatment.

One limitation of our study is that the search strategy was restricted to the PubMed/MEDLINE and ClinicalTrials.gov databases. As a result, we may have missed some eligible studies. However, PubMed is one of the largest biomedical bibliographic databases, and ClinicalTrials.gov is a key reference database for interventional trials.

## 5. Conclusions

Recent advances in pharmaceutical technology have enabled the development of orally administered SRLs, providing a promising alternative to iSRLs in the management of acromegaly. OOCs have demonstrated efficacy in maintaining long-term biochemical control in patients previously stabilized on iSRLs. paltusotine has shown the ability to sustain biochemical control, but its long-term efficacy has yet to be conclusively established.

## Figures and Tables

**Figure 1 pharmaceutics-16-01357-f001:**
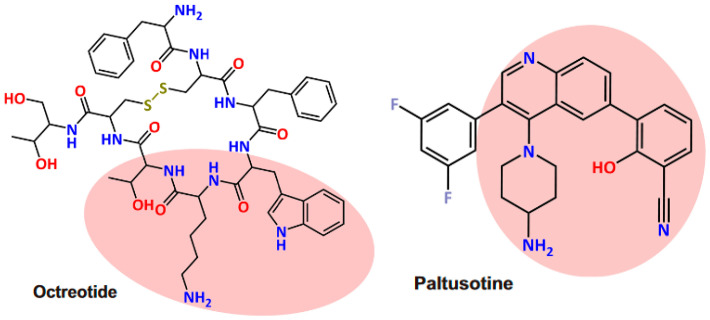
Chemical structures of the somatostatin receptor ligands octreotide and paltusotine. Pink area highlights similar moieties. Reproduced from [16] under a Creative Commons Attribution 4.0 International License (http://creativecommons.org/licenses/by/4.0/).

**Figure 2 pharmaceutics-16-01357-f002:**
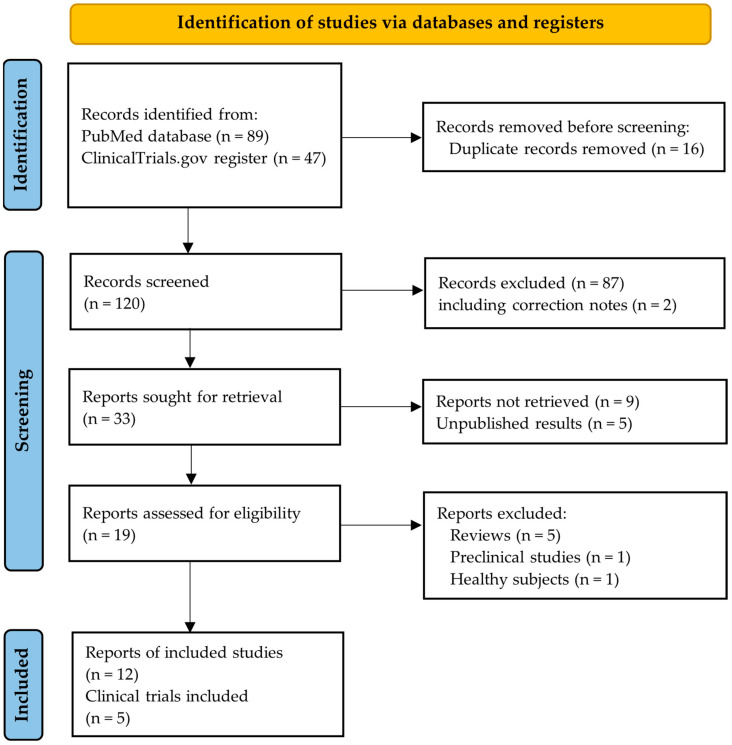
PRISMA flowchart.

**Table 1 pharmaceutics-16-01357-t001:** Inclusion and exclusion criteria for the selection of studies.

PICOS Parameter	Inclusion Criteria	Exclusion Criteria
Population	Adult individuals with confirmed acromegaly. Any study population size, gender, disease duration, prior treatment (surgery, radiotherapy, none), and degree of clinical and biochemical control.	Non-human animals. Healthy individuals. Patients with syndromic acromegaly: McCune Albright syndrome (OMIM ^1^: #174800), Carney syndrome (#160980), X-linked acrogigantism (#300942).
Intervention	Oral administration of SRLs (octreotide, lanreotide, pasireotide, and paltusotine). Any duration, dose, and administration frequency.	Treatment with a drug other than SRLs in order to treat acromegaly; in particular, dopamine and GH receptor agonists.
Comparison	With or without a comparison group (placebo, iSRLs, other drug, radiotherapy, surgery).	
Outcomes	Proportion of responders at the end of the intervention based on serum IGF-1 and/or GH levels (any threshold).	
Study design	Any design (open-label, double blind) Records published in English. Results published in peer-review journals.	Reviews, meta-analysis, case reports, case series, commentaries.

^1^ OMIM: Online Mendelian Inheritance in Man.

**Table 2 pharmaceutics-16-01357-t002:** Characteristics of the selected studies.

Ref., Publication Year	Trial Name (ClinicalTrials ID) Sponsor	Study Design	Study Population	Intervention
[12,26], 2015	CH-ACM-01 (NCT01412424) Chiasma™	Phase III trial Single-arm Open-label	151 patients with acromegaly and IGF-1 < 1.3 ULN and 2 h integrated GH < 2.5 ng/mL receiving iSRL for ≥3 months.	Core phase (7 months): Administration of the first OOC dose ≥ 4 weeks after the last SRLs injection. OOC 40 mg, escalation to 60, and then 80 mg/d if there was inadequate IGF-1 suppression (2–5 months) followed by a fixed-dose period up to the end of the core phase. Extension phase (6 months): At the fixed-dose determined during the core phase.
[13,27], 2020	OPTIMAL (NCT03252353) Chiasma™	Phase III trial Placebo-controlled Double-blind	56 patients with acromegaly and IGF-1 ≤ 1.0 × ULN after ≥6 months on long-acting iSRL.	Core phase (9 months): OOC (*n* = 28) or placebo (*n* = 28) twice daily (administration of the first dose ≥4 weeks after the last SRLs injection). OOC 40 mg/d, escalation to 60, and then 80 mg/d if there was inadequate IGF-1 (increase ≥ 30% to >1.0 × ULN or >1.0 × ULN for 2 consecutive visits) and/or symptoms suppression.
[28], 2022		Phase III trial Single-arm Open-label	40 patients with completion of the 9-month placebo-controlled phase (20 from the OOC group and 20 from the placebo group).	Extension phase (11 months): OOC 60 mg/d followed by a subsequent dose escalation (80 mg/d, 67.5%) or de-escalation (40 mg/d, 7.5%) based on biochemical results, safety and disease symptoms.
[29,30], 2021	MPOWERED (NCT02685709) Chiasma™	Phase III trial iSRL-controlled Open-label	146 patients with acromegaly and IGF-1 < 1.3 × ULN and mean integrated GH < 2.5 ng/mL on long-acting iSRL for ≥6 months. 92 patients with IGF-1 < 1.3 × ULN and mean integrated GH < 2.5 ng/mL at the end of the run-in phase.	Run-in phase (6 months): OOC 40 mg/d, escalation to 60 or 80 mg/d based on IGF-1 results and acromegaly symptoms. Randomized treatment phase (8.3 months): OOC at the final dose achieved during the run-in-phase (*n* = 55) or iSRL at the dose and interval before trial enrolment (*n* = 37).
[31], 2023		Phase III trial Single-arm Open-label	60 patients with acromegaly and IGF-1 < 1.3 × ULN after the core trial.	Extension phase (≤3.5 years): OOC at the final dose achieved during the run-in-phase.
[32,33], 2022	ACROBAT Edge (NCT03789656) Crinetics Pharmaceuticals™	Phase II trial Single-arm Open-label	47 patients with acromegaly with stable treatment for ≥3 months (including a group of 25 partial responders previously treated with iSRL with IGF-1 ≤ 2.5 × ULN, and a group of 15 partial responders previously treated with iSRL plus dopamine agonist).	Treatment phase (13 weeks): Paltusotine 10 mg/d, escalation to 20, 30, and 40 mg/d. Washout phase (1 month): No treatment
[15,34], 2024	PATHFNDR-1 (NCT04837040) Crinetics Pharmaceuticals™	Phase III trial Placebo-controlled Double-blind	58 patients with acromegaly and IGF-1 ≤ 1.0 × ULN on iSRL for at least 12 weeks.	Core phase (8.3 months): Paltusotine (*n* = 30) or placebo (*n* = 28). Paltusotine 40 mg 1x/d, escalation to 60 mg/d if IGF-1 > 0.9 ULN.

**Table 3 pharmaceutics-16-01357-t003:** Characteristics of participants within the studies included in the review.

Ref.	Trial	Inclusion and Exclusion Criteria
[12]	CH-ACM-01	Inclusion: Adult subjects between 18 and 75 years; patients with acromegaly and documented evidence of a GH-secreting pituitary tumor that is abnormally responsive to glucose, or documented elevated IGF-1, who are currently receiving a stable dose of iSRL for at least the previous 3 months; a serum IGF-1 level < 1.3 × ULN and a serum-integrated GH response over 2 h < 2.5 ng/mL. Exclusion: GH antagonist (within <3 months) or dopamine agonist (within <2 months); radiotherapy within 10 years; pituitary surgery within 6 months before screening; conditions (e.g., bariatric surgery) significantly affecting gastric acidity or emptying; Current use (within 1 month) of proton pump inhibitors and current chronic use of H2-antagonists; female patients who are pregnant or lactating.
[13]	OPTIMAL	Inclusion: Adult subjects ≥ 18 years; confirmed acromegaly: pituitary tumor on imaging or pathology analysis and IGF-1 ≥ 1.3 ULN; IGF-1 ≤ 1.0 × ULN (based on the average of two assessments); treatment with long-acting iSRLs (octreotide or lanreotide) ≥6 months with a stable dose for ≥3 months. Exclusion: Treatment with iSRLs not as indicated in the label; pituitary surgery within 6 months; pituitary radiotherapy; participation in the CH-ACM-01 or MPOWERED trials; symptomatic cholelithiasis; treatment with pegvisomant within 24 weeks, dopamine agonists within 12 weeks, or pasireotide within 24 weeks.
[29]	MPOWERED	Inclusion: Confirmed diagnosis of acromegaly; biochemical control at screening (IGF-1 < 1.3 × ULN and mean integrated GH < 2.5 ng/mL over 2 h); treatment with long-acting forms of iSRL for ≥6 months. Exclusion: Injections of iSRLs at a dosing interval >8 weeks; pituitary radiotherapy within 5 years; pituitary surgery within 6 months; previous participation in the CH-ACM-01 trial; any clinically significant uncontrolled concomitant disease; symptomatic cholelithiasis; treatment with pegvisomant (within 12 weeks), dopamine agonists (within 6 weeks), or pasireotide (within 12 weeks).
[32]	ACROBAT Edge	Inclusion: Age between 18 and 70 years; confirmed diagnosis of acromegaly with either a partial or complete response with SRLs. Exclusion: Treatment-naïve acromegaly subjects; prior treatment with paltusotine; pituitary surgery within 6 months prior to screening; history or presence of malignancy except adequately treated basal cell and squamous cell carcinomas of the skin within the past 5 years; use of any investigational drug within the past 30 days or 5 half-lives; positive screening for human immunodeficiency virus, hepatitis B surface antigen, or hepatitis C antibody or has a history of a positive result; history of alcohol or substance abuse in the past year; any condition that, in the opinion of the investigator, would jeopardize the subject’s appropriate participation in this study; cardiovascular conditions or medications associated with prolonged QT or those which predispose subjects to heart rhythm abnormalities; subjects with symptomatic cholelithiasis; subjects with clinically significant abnormal findings during the screening period, and any other medical condition(s) or laboratory findings that might jeopardize the subject’s safety or ability to complete the study; subjects taking injectable octreotide at a dose higher than 40 mg, or lanreotide depot at a dose higher than 120 mg, or pasireotide LAR at a dose higher than 60 mg; patients who usually take injectable octreotide or lanreotide depot less frequently than every 4 weeks.
[15]	PATHFNDR-1	Inclusion: Age ≥ 18 years; confirmed diagnosis of acromegaly; biochemical control as measured by IGF-1 ≤ 1.0 × ULN via stable dose of iSRL therapy; females either surgically sterile, post-menopausal, or using an effective method of birth control. Exclusion: Treatment-naïve or treatment-withdrawn acromegaly subjects; prior treatment with paltusotine; pituitary surgery within 24 weeks prior to screening or history of pituitary radiation therapy; history or presence of malignancy except adequately treated basal cell and squamous cell carcinomas of the skin within the past 5 years; use of any investigational drug within the past 30 days or 5 half-lives, whichever is longer; history of HIV, hepatitis B, or active hepatitis C; pregnancy; history of alcohol or substance abuse in the past 12 months; any condition that, in the opinion of the investigator, would jeopardize the subject’s appropriate participation in this study; cardiovascular conditions or medications associated with prolonged QT or those which predispose subjects to heart rhythm abnormalities; subjects with symptomatic cholelithiasis; subjects with clinically significant abnormal findings during the screening period, or any other medical condition(s) or laboratory findings that, in the opinion of the Investigator, might jeopardize the subject’s safety or ability to complete the study; subjects currently taking long-acting pasireotide (within 24 weeks prior to screening) or pegvisomant, dopamine agonists (within 12 weeks prior to screening), or short-acting somatostatin analogs (within 12 weeks prior to first dose of study drug).

**Table 4 pharmaceutics-16-01357-t004:** Assessment of the quality of the randomized interventional trials. Green indicates a low risk of bias, while orange signifies some concerns regarding the risk of bias.

	Randomization Process	Deviations from Intended Interventions	Missing Data	Measurement of the Outcome	Selection of the Results	Overall Judgment
OPTIMAL (core phase)						
MPOWERED (core phase)						
PATHFNDR-1 (core phase)						

**Table 5 pharmaceutics-16-01357-t005:** Assessment of the quality of the uncontrolled interventional trials. Green indicates a low risk of bias, while orange signifies some concerns regarding the risk of bias.

	Confounding	Selection Bias	Missing Data	Measurement of the Outcome	Selection of the Results	Overall Judgment
CH-ACM-01 (core and extension phases)						
ACROBAT Edge (treatment phase)						

**Table 6 pharmaceutics-16-01357-t006:** Efficacy of oral SRLs based on biochemical response.

Ref.	Trial Sponsor	Biochemical Response
[12]	CH-ACM-01	From baseline to the end of the core phase (OOC for 7 months) (*n* = 151): IGF-1 < 1.3 × ULN: from 91.4% (85.7–95.3%) to 64.2% (56.0–71.9%). IGF-1 < 1.3 × ULN + mean GH < 2.5 ng/mL: from 88.7% (82.6–93.3%) to 61.6% (53.3–69.4%). IGF-1 ≤ 1.0 × ULN: from 63.6% (55.4–71.3%) to 37.8% (30.0–46.0%). IGF-1 ≤ 1.0 × ULN + mean GH < 1.0 ng/mL: from 43.1% (35.0–51.4%) to 32.5% (25.1–40.5%). From baseline to the end of the extension phase (OOC up to 13 months) (*n* = 110): IGF-1 < 1.3 × ULN + mean integrated GH < 2.5 ng/mL: from 82.7% (74.4–89.3%) to 74.6% (65.4–82.4%). IGF-1 < 1.3 × ULN: from 82.7% (74.4–89.3%) to 76.4% (67.3–83.9%). IGF-1 ≤ 1.0 × ULN: from 53.6% (43.9–63.2%) to 47.3% (37.7–57.0%).
[13,28]	OPTIMAL	From baseline to the end of the core phase (9 months) (*n* = 56): IGF-1 ≤ 1.0 × ULN: - OOC: from 96.4% (81.7–99.9%) to 58.2% (37.4–75.8%). - Placebo: from 82.1% (63.1–93.9%) to 19.4% (7.4–38.2%). - *p* = 0.008 for the comparison between the two groups at the end of the treatment. GH < 2.5 ng/mL: - OOC: from 96.4% (81.7–99.9%) to 77.7% (57.6–91.8%). - Placebo: from 89.3% (71.8–97.7%) to 30.4% (15.0–50.5%). - *p* < 0.001 for the comparison between the two groups at the end of the treatment. IGF-1 < 1.3 × ULN: - OOC: from 100.0% (87.8–100.0%) to 71.4% (51.3–86.8%). - Placebo: data not available. From baseline to the end of the open-label extension phase (11 months) (*n* = 40): IGF-1 ≤ 1.0 × ULN: 70.0% (53.5–83.4%) at baseline - From 73.7% (48.8–90.9%) to 52.6% (28.9–75.6%) in patients who received OOC during the core phase, but with a responder rate of 92.6% (78.7–100%) when accounting for missing IGF-1 data.
[29,31]	MPOWERED	From baseline to the end of the randomized treatment phase (*n* = 92): IGF-1 < 1.3 × ULN (time-weighted average): - OOC: 89.1% (77.8–95.3%) to 90.9% (80.0–96.5%). - iSRL: 100.0% (90.5–100.0%) to 100.0% (90.5–100.0%). - Statistical non-inferiority of OOC compared to iSRL. Annual evaluation during the extension phase (≤3.5 years) (*n* = 60): IGF-1 ≤ 1.3 × ULN: - Year 1: from 100% (94.0–100.0%) to 89.7% (78.8–96.1) at the end of year 1 (*n* = 58). - Year 2: 87.8% (73.8–95.9) at the end of year 2 (*n* = 41). - Year 3: 93.5% (78.6–99.2) at the end of year 3 (*n* = 31).
[32]	ACROBAT Edge	From baseline to the end of the treatment phase (13 weeks): IGF-1 levels: - Patients previously on iSRL monotherapy (*n* = 25): from 1.34 (1.08–1.47) to 1.34 (1.17–1.45) × ULN (*p* = 0.63). - Patients previously treated with iSRL plus dopamine agonist (*n* = 15): from 1.21 × ULN (0.93–1.50) to 1.44 × ULN (1.09–2.08) (*p* = 0.002). GH levels: - Patients previously on iSRL monotherapy: from 0.69 ng/mL (0.49–1.55) to 0.72 ng/mL (0.48–1.75) (*p* = 0.63). - Patients previously treated with iSRL plus dopamine agonist: from 1.04 ng/mL (0.51–1.61) to 1.46 ng/mL (0.62–3.65) (*p* = 0.0067).
[15]	PATHFNDR-1	From baseline to the end of the core phase (8.3 months) (*n* = 58): IGF-1 ≤ 1.0 × ULN: - Paltusotine: from 100.0% (88.4–100.0%) to 83.3% (65.3–94.4%). - Placebo: from 100.0% (87.7–100.0%) to 3.6% (0.1–18.4%). - *p* < 0.0001 between the two groups. Mean GH < 1.0 ng/mL (*n* = 41): - Paltusotine (*n* = 23): from 100.0% (85.2–100.0%) to 87.0% (66.4–97.2%). - Placebo (*n* = 18): from 100.0% (81.5–100.0%) to 27.8% (9.7–53.5%). - *p* = 0.0003 between the two groups.

Data are percentage of responders (95% confidence interval) or mean (interquartile range), as appropriate.

## Data Availability

The raw data supporting the conclusions of this article will be made available by the authors on request.

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
