# Peer review of "Advances in the Oral Administration of Somatostatin Receptor Ligands in Acromegaly: A Systematic Review Focusing on Biochemical Response"

_pharmaceutics, 2024, doi:10.3390/pharmaceutics16111357_

Round 1
Reviewer 1 Report
Comments and Suggestions for Authors
A systematic literature review is presented on advances in the oral administration of somatostatin receptor ligands, which are a mainstay of therapy because they reduce growth hormone secretion. The current challenges of oral delivery are presented, given the low permeability and the need for a less traumatic and painful delivery system such as injectables.
The results are discussed in terms of biochemical response.
The topic is of great interest, the document is well written and presents relevant information. I propose that it be accepted for publication after a few minor corrections.
1. The introduction should include global statistics on the prevalence of acromegaly.
2. It is important to justify the databases consulted; one might think that the search could have been more extensive and rigorous.
3. Rotating the words Identification, Screening, and Included 90 degrees would improve Figure 2.
4. The color nomenclature in Tables 4-5 should be presented and explained.
5. The paragraph between lines 215 and 220 is a good introduction, but given the nature of the manuscript, it is always important to justify each claim. Why is this an active line of research? Only 31 sources of information were consulted, of which 13 were published between 2022 and 2023, and there is only one from 2024. Describe what is the promising progress
Author Response
Responses to the Reviewer 1
A systematic literature review is presented on advances in the oral administration of somatostatin receptor ligands, which are a mainstay of therapy because they reduce growth hormone secretion. The current challenges of oral delivery are presented, given the low permeability and the need for a less traumatic and painful delivery system such as injectables.
The results are discussed in terms of biochemical response.
The topic is of great interest, the document is well written and presents relevant information. I propose that it be accepted for publication after a few minor corrections.
We are very grateful to the Reviewer for his thoughtful comments, as well as for the time dedicated to the review process. All changes are in red in the revised manuscript.
Comment 1. The introduction should include global statistics on the prevalence of acromegaly.
We agree with the Reviewer that including global statistics on the prevalence of acromegaly strengthens the introduction of the topic. Consequently, we have incorporated this point on global epidemiology as follows (page 1, lines 32-34) : « The prevalence of acromegaly is estimated at 5.9 cases per 100,000 individuals, with an incidence rate of approximately 0.38 cases per 100,000 person-years [1]. »
Comment 2. It is important to justify the databases consulted; one might think that the search could have been more extensive and rigorous.
We conducted a search using the PubMed/MEDLINE database, one of the largest biomedical databases, and the ClinicalTrials.gov database, a key reference for registered clinical trials. However, we acknowledge the Reviewer's comment that broader searches across additional databases could increase the identification of relevant studies. We have addressed this limitation in the Discussion section (page 12, lines 274-277): ”One limitation of our study is that the search strategy was restricted to the PubMed/MEDLINE and ClinicalTrials.gov databases. As a result, we may have missed some eligible studies. However, PubMed is one of the largest biomedical bibliographic databases, and ClinicalTrials.gov is a key reference database for interventional trials.”.
Comment 3. Rotating the words Identification, Screening, and Included 90 degrees would improve Figure 2.
We thank the Reviewer for identifying this typo, which occurred during the conversion of the Word file to PDF. We have corrected this by converting the figure to a .png file (page 5, line 166).
Comment 4. The color nomenclature in Tables 4-5 should be presented and explained.
We thank the Reviewer for highlighting this lack of clarity of the Tables 4 and 5. Therefore, we have added the explanation on the colors used in Tables 4 & 5, as follows (page 9, lines 190-193): “Green indicates a low risk of bias, while orange signifies some concerns regarding the risk of bias.”.
Comment 5. The paragraph between lines 215 and 220 is a good introduction, but given the nature of the manuscript, it is always important to justify each claim. Why is this an active line of research? Only 31 sources of information were consulted, of which 13 were published between 2022 and 2023, and there is only one from 2024. Describe what is the promising progress
As suggested by the Reviewer, we have improved the first paragraph of the Discussion section as follows (page 11, lines 220-231): “The oral administration of SRLs, such as OOC and oral paltusotine, represents an active area of research aiming to provide an efficient alternative to iSRLs in the management of acromegaly. It represents a genuine hope to improve the quality of life for patients with acromegaly. This development represents a promising step forward in the management of acromegaly, offering an alternative to the limitations associated with iSRLs. In particular, iSRLs exhibit disadvantages such injection-site pain, nodules, swelling, bruising, inflammation, and scarring. Additionally, some patients may experience worsening symptoms of acromegaly and higher IGF-1 levels close to the date of their next injection [5,35]. ”.

Reviewer 2 Report
Comments and Suggestions for Authors
The manuscript is a significant contribution to the field of Acromegaly treatment. This review focused on oral somatostatin receptor ligands (SRLs) in controlling acromegaly based on biochemical response. The authors have nicely presented the literature reports of SRLs. The manuscript is well-written and summarized. This reviewer has some suggestions.
1. Authors carefully showed the clinical outcomes for oral somatostatin receptor ligands (SRLs), stating very minor disadvantages of injectable SRLs. Authors are asked to discuss the disadvantages of injectable SRLs in detail.
2. Authors showed clinical benefits of oral SRLs over injectable SRLs. Please discuss the pharmacokinetics and other ADMET properties in more detail to show the benefits.
3. The authors briefly discussed the biochemical responses for oral SRLs in different clinical trials. Please include some preclinical data of the oral drugs especially for paltusotine for better understanding.
Author Response
Responses to the Reviewer 2
The manuscript is a significant contribution to the field of Acromegaly treatment. This review focused on oral somatostatin receptor ligands (SRLs) in controlling acromegaly based on biochemical response. The authors have nicely presented the literature reports of SRLs. The manuscript is well-written and summarized. This reviewer has some suggestions.
We are very grateful to the Reviewer for his thoughtful comments, as well as for the time dedicated to the review process. All changes are in red in the revised manuscript.
- Authors carefully showed the clinical outcomes for oral somatostatin receptor ligands (SRLs), stating very minor disadvantages of injectable SRLs. Authors are asked to discuss the disadvantages of injectable SRLs in detail.
As suggested by the Reviewer, we now discuss the disadvantages of injectable SRLs in a more detailed way. Therefore, we have added some sentences on this topic in the revised Discussion section (page 11, lines 214-222): “The oral administration of SRLs, such as OOC and oral paltusotine, represents an active area of research aimed at improving the quality of life for patients with acromegaly. This development represents a promising step forward in the management of acromegaly, offering an alternative to the limitations associated with iSRLs. In particular, iSRLs exhibit disadvantages such injection-site pain, nodules, swelling, bruising, inflammation, and scarring. Additionally, some patients may experience worsening symptoms of acromegaly and higher IGF-1 levels close to the date of their next injection [33,34]. However, the transition from injectable to oral SRLs should not compromise disease control. Our review indicates that the current data is reassuring in this regard.”.
- Authors showed clinical benefits of oral SRLs over injectable SRLs. Please discuss the pharmacokinetics and other ADMET properties in more detail to show the benefits.
As suggested by the Reviewer, we now discuss the pharmacokinetics in a more detailed way:
- For oral octreotide (page 11, lines 229-234): “The most studied formulation is OOC, administered twice daily in fasting state. The pharmacokinetic parameters in human subjects are similar after oral administration of 20 mg octreotide or subcutaneous administration of 100 µg octreotide, in particular regarding the peak plasma concentration and the area under the curve [9]. The mean of apparent steady state elimination half-life ranged from 3.19 ± 1.07 h on 40 mg of oral octreotide, to 4.47 ± 2.02 h on 80 mg [10].”.
- For oral paltusotine (page 11, lines 246-248): “In contrast to OOC, paltusotine offers the advantage of once-daily administration. Paltusotine is associated with increased plasma concentrations to doses up to 40 mg, and is eliminated with a half-life of approximately 30 h [35].”.
- The authors briefly discussed the biochemical responses for oral SRLs in different clinical trials. Please include some preclinical data of the oral drugs especially for paltusotine for better understanding.
As suggested by the Reviewer, we have included some preclinical data on paltusotine in the revised Discussion section, as follows (page 11, lines 248-251): “Preclinical studies had demonstrated that paltusotine, an agonist of somatostatin receptor subtype 2, dose dependently suppresses recombinant GH-releasing hormone-stimulated GH secretion in rats, and dose dependently reduces IGF-1 levels [36]. ».
Reviewer 3 Report
Comments and Suggestions for Authors
The manuscript titled "Advances in Oral Administration of Somatostatin Receptor Ligands in Acromegaly: A Review focusing on Biochemical Response." by Clémence Reverdiau and Damien Denimal renders a very comprehensive overview of the efficacy of oral somatostatin receptor ligands, namely octreotide capsules and paltusotine. It is very enlightening, informative, and well written. However, there are a few minor details that (in my opinion) need to be addressed:
* In the introduction, disadvantages of injectable somatostatin receptor ligands are enumerated and backed by data from a quite recent study. Given that most respondents in that study (O’Toole et al. 2023; ref. 2) had NET, it seems a little out of place considerung that there are data from a real-world PRO study in 195 patients with acromegaly (Strasburger et al. EJE 2016; 10.1530/EJE-15-1042). In this study, an astounding 70% of patients reported site pain lasting up to a week following injection.
Other great papers in this regard are Geer et al. BMC Endocr Disord 2020 (10.1186/s12902-020-00595-4), which also includes more acromegaly patients than O'Toole et al. (but clearly less than Straburger et al.). In this context, it might also be worth mentioning, that there is a discrepancy between patient-reported outcomes in terms of injection site reactions and estimations by providers of medical care (cf. Geer et al. Pituitary 2020; 10.1007/s11102-019-01013-2).
Delving deeper, there is another paper that should be brought to attention: Sisco et al. Curr Med Res Opin 2024 (10.1080/03007995.2024.2314244) is a study using an online choice experiment in 109 adults with acromegaly investigating preferences for different treatment options. Surprisingly, given the choice between subcutaneous injections with a pen at home over oral octreotide capsules, 65% opted for the injection. This is counter-intuitive, but it reflects a sentiment of patients from the OOC studies that found the twice-daily routine requiring specific fasting conditions to be burdensome, let alone the cumbersome to swallow capsules. So, after all it is probably not that surprising. (Admittedly though, the study was funded by Camurus, so we cannot rule out bias and need to be cautious with these data.)
* In line 102f. It should read ClinicalTrials.gov
* Line 197f.: „Interestingly, …“ I’m wondering why it is interesting that the efficacy of an investigational drug is higher than placebo. That’s what I’d expect from any [reasonably pre-tested] investigational product. And: What does +62.7% mean in this context?
* Line 204: IGF-1 instead of IFG-1
Overall, this is a wonderful piece of work and deserves publication after minor revision of the above-mentioned issues.
Author Response
The manuscript titled "Advances in Oral Administration of Somatostatin Receptor Ligands in Acromegaly: A Review focusing on Biochemical Response." by Clémence Reverdiau and Damien Denimal renders a very comprehensive overview of the efficacy of oral somatostatin receptor ligands, namely octreotide capsules and paltusotine. It is very enlightening, informative, and well written. However, there are a few minor details that (in my opinion) need to be addressed:
We are very grateful to the Reviewer for his thoughtful comments, as well as for the time dedicated to the review process. All changes are in red in the revised manuscript.
* In the introduction, disadvantages of injectable somatostatin receptor ligands are enumerated and backed by data from a quite recent study. Given that most respondents in that study (O’Toole et al. 2023; ref. 2) had NET, it seems a little out of place considering that there are data from a real-world PRO study in 195 patients with acromegaly (Strasburger et al. EJE 2016; 10.1530/EJE-15-1042). In this study, an astounding 70% of patients reported site pain lasting up to a week following injection.
Other great papers in this regard are Geer et al. BMC Endocr Disord 2020 (10.1186/s12902-020-00595-4), which also includes more acromegaly patients than O'Toole et al. (but clearly less than Straburger et al.). In this context, it might also be worth mentioning, that there is a discrepancy between patient-reported outcomes in terms of injection site reactions and estimations by providers of medical care (cf. Geer et al. Pituitary 2020; 10.1007/s11102-019-01013-2).
Delving deeper, there is another paper that should be brought to attention: Sisco et al. Curr Med Res Opin 2024 (10.1080/03007995.2024.2314244) is a study using an online choice experiment in 109 adults with acromegaly investigating preferences for different treatment options. Surprisingly, given the choice between subcutaneous injections with a pen at home over oral octreotide capsules, 65% opted for the injection. This is counter-intuitive, but it reflects a sentiment of patients from the OOC studies that found the twice-daily routine requiring specific fasting conditions to be burdensome, let alone the cumbersome to swallow capsules. So, after all it is probably not that surprising. (Admittedly though, the study was funded by Camurus, so we cannot rule out bias and need to be cautious with these data.)
We thank the Reviewer for highlighting these references. As suggested by the Reviewer, we have added the following references:
- Straburger et al. (EJE 2016) and Geer et al. (BMC Endocrin Disord 2020): we now mention these two papers in the Introduction section (page 2, lines 44-50): “Injectable SRLs (iSRLs) exhibit numerous disadvantages such injection-site pain, nodules, bruising, inflammation, and scarring [3–5]. A real-world survey showed that 70% of patients reported experiencing pain at the injection site lasting up to a week, with other common adverse effects including nodules (38%), swelling (28%), bruising (16%), scar tissue formation (8%), and inflammation (7%) [3]. In addition, nurses have only a little bit confidence that octreotide syringe will not be clogged and somewhat easy to use during injection [6].”.
- Sisco et al. (Curr Med Res Opin 2024): following the concern raised by the Reviewer, we now discuss this counter-intuitive result provided by this survey (page 11, lines 228-231): “However, a recent survey revealed that 65% of patients preferred subcutaneous injections administered once every fourth weeks using a pen at home, compared to oral capsules taken twice daily, suggesting that required three-hour fasting period remains a significant burden [36].”.
* In line 102f. It should read ClinicalTrials.gov
We thank the Reviewer for having detected this typo. We have corrected this typo as follows (page 3, lines 102-103): “We conducted a search procedure using the PubMed/MEDLINE and ClinicalTrials.gov databases.”.
* Line 197f.: „Interestingly, …“ I’m wondering why it is interesting that the efficacy of an investigational drug is higher than placebo. That’s what I’d expect from any [reasonably pre-tested] investigational product. And: What does +62.7% mean in this context?
The OPTIMAL study provides the first evidence from a randomized, prospective trial in humans demonstrating that OOC is superior to placebo for disease control. We agree with the Reviewer that it is what we would expect in this type of trial. In addition, the term “interestingly” is a subjective judgment that is not appropriate for the Results section. Therefore, we have removed the word “interestingly” (page 9, lines 200-201)."
* Line 204: IGF-1 instead of IFG-1
We thank the Reviewer for having detected this typo. We have corrected this typo as follows (page 9, lines 207-208): “In the ACROBAT Edge trial, IGF-1 levels were expressed in concentrations rather than in percentage of responders.”.
Overall, this is a wonderful piece of work and deserves publication after minor revision of the above-mentioned issues.